# Pulse Broadening Effects on Ranging Performance of a Laser Altimeter with Return-to-Zero Pseudorandom Noise Code Modulation

**DOI:** 10.3390/s22093293

**Published:** 2022-04-25

**Authors:** Hyung-Chul Lim, Jong Uk Park, Mansoo Choi, Eunseo Park, Ki-Pyoung Sung, Jung Hyun Jo

**Affiliations:** Korea Astronomy and Space Science Institute, Daejeon 34055, Korea; jupark@kasi.re.kr (J.U.P.); cmsoo@kasi.re.kr (M.C.); skel93@kasi.re.kr (E.P.); kpsung@kasi.re.kr (K.-P.S.); jhjo39@kasi.re.kr (J.H.J.)

**Keywords:** laser altimeter, pseudorandom noise code, pulse broadening effect, flat-topped multi-Gaussian beam, cross-correlation

## Abstract

A laser altimeter using code modulation techniques receives a backscattered pulse wider than the transmitted rectangular pulse when scanning a rough or sloped target surface. This leads to degrading the ranging performance in terms of signal-to-noise ratio (SNR) and detection probability. Unlike the pulsed techniques, little work has focused on the pulse broadening effect of the code modulation techniques. In this study, mathematical models were derived to investigate the pulse broadening effect on the ranging performance of a return-to-zero pseudorandom noise (RZPN) laser altimeter. Considering that the impulse response can be approximated by a Gaussian function, the analytical waveform was derived using a new flat-topped multi-Gaussian beam (FMGB) model. The closed-form expressions were also analytically derived for a peak cross-correlation, SNR, and detection probability in terms of the pulse broadening effect. With the use of a three-dimensional model of asteroid Itokawa for practical surface profiles, the analytical expressions were validated by comparing to the results obtained from numerical simulations. It was also demonstrated that the pulse broadening effect dropped down the peak cross-correlation and then deteriorated the ranging performance. These analytical expressions will play an important role in not only designing a laser altimeter using the RZPN code modulation technique but also analyzing its ranging performance.

## 1. Introduction

Laser altimeters have become essential instruments in many lunar, planetary, and asteroid exploration missions since their first use onboard the Apollo 15 mission [1]. They provide not only the topographic map by measuring the distance to the target surface, but also terrain surface characteristics of slope, roughness, and reflectivity by analyzing received waveforms. They also play an important role as navigation sensors in the rendezvous and touchdown phase as well as long range measurement. Conventional laser altimeters use the pulsed time-of-flight (TOF) technique which allows range measurements from several meters to hundreds of kilometers and provide high range precision at a decimeter or even centimeter level by employing short laser pulses [2]. After the first three laser altimeters used for Apollo 15, 16, and 17 missions, all subsequent laser altimeters have used diode pumped Q-switching Nd:YAG lasers [3,4]. The pulsed laser altimeters require high peak power laser pulses (i.e., high pulse energy from a few to tens of millijoules) to increase single shot detection probability. Consequently, the laser tends to be bulky, heavy, and leads to low optical conversion efficiency because the transmitted energy is concentrated in the small pulse width of several nanoseconds to achieve high pulse energy.

Due to the size, weight, and power (SWaP) requirements of spacecrafts, laser altimeters must be compact, robust, and power-efficient, keeping a long lifetime during the space mission. The pseudorandom noise (PN) code modulation technique [5,6] is considered a promising alternative to achieve the SWaP requirements for space-based applications. It permits the use of a compact, power-efficient, eye-safe, less expensive, and commercially available continuous wave (CW) diode laser [7]. The PN laser altimeter transmits the laser pulse sequence modulated with PN codes on a CW laser and receives the pulse trains backscattered by the target surface. It then determines the distance by a peak search of the cross-correlation in the time domain.

Compared to the pulsed TOF technique under the same average transmitted power, the PN code modulation technique provides low signal-to-noise ratio (SNR) which imposes restrictions on space-based applications, even though it has been widely used for atmospheric remote sensing [8,9,10,11,12]. However, the low SNR problem can be overcome by not only using a CW laser with high average power but also accumulating the received pulse trains into a histogram. Recent progress provides high average power by employing the master oscillator fiber amplifier (MOPA) architecture consisting of a master laser and fiber amplifiers to boost the output power. The CW lasers based on the MOPA architecture, providing high average power and small SWaP, have been demonstrated for space laser communications [13,14,15,16]. The received pulse trains can be accumulated into a histogram without changing the peak location of the cross-correlation, which increases SNR in proportion to the square root of the accumulation number and then improves the detection probability [5,6,17]. In addition, SNR can be improved by using state-of-the-art HgCdTe avalanche photodiodes (APDs) which allow single photon sensitivity to provide high linear gain, high quantum efficiency, and low dark rate [18,19,20]. The PN code modulation technique has been proposed for some space applications [3,21,22] and was also applied to the LISA mission to detect gravitational waves in space with three satellites separated by five million kilometers on a heliocentric orbit [23]. In addition, the return-to-zero PN (RZPN) code modulation technique was proposed to reduce background noise over the integration time (i.e., accumulation time to make a histogram), thereby improving SNR [24,25,26].

In laser altimetry, the received waveform or the impulse response is approximated by a Gaussian function if the transmitted beam has a Gaussian shape in terms of temporal and spatial domains [27,28,29,30,31]. It is also modeled as a series of Gaussian functions which is decomposed to retrieve target surface characteristics such as slope and roughness [32,33,34,35]. However, the received waveform is distorted and broadened by the target surface profile and beam divergence. The pulse broadening effect degrades the ranging performance of laser altimeters using code modulation techniques because the received waveform spreads out more widely than the transmitted rectangular pulse width which results in the reduction of SNR. Thus, the effect should be considered when evaluating the figures of merit of laser altimeters based on code modulation techniques (i.e., PN and RZPN techniques). The pulse broadening effect has been studied in many laser altimeters with the pulsed TOF technique [27,28,29,30,31], but little work has been done for those using code modulation techniques.

The transmitted rectangular pulse can be analytically expressed by the flat-topped multi-Gaussian beam (FMGB) consisting of a finite sum of Gaussian beams [36,37], and its received waveform can be also described by a finite sum of pulse broadened Gaussian beams. Unlike the RZPN technique, the PN technique cannot consider the pulse broadening effect because the backscattered photons by the target surface are spread not just in a single bit duration corresponding to the transmitted rectangular pulse, but also in the neighboring bits. In this study, we provide the detection models of a laser altimeter with the RZPN code modulation technique by considering the pulse broadening effect. Here, we propose a new FMGB not only to describe the transmitted rectangular pulse and received waveform consisting of Gaussian beams, but also to derive the cross-correlation between the code kernel and the received waveform. Mathematical models were also developed to investigate the pulse broadening effect on the ranging performance of an RZPN laser altimeter in terms of SNR and detection probability. Using asteroid Itokawa as a target surface for more practical investigation, numerical simulations were performed to evaluate the detection models and analyze the pulse broadening effect. It is demonstrated that the mathematical detection models are practical, and the pulse broadening effect significantly deteriorates the ranging performance by decreasing the cross-correlation.

## 2. Detection Models of RZPN Laser Altimeter

### 2.1. Flat-Topped Multi-Gaussian Beam

Flat-topped laser beams are gaining more and more attention due to their wide use in inertial confinement fusion, optical communications, material processing, electron acceleration, and nonlinear optics [37], which has a uniform intensity profile spatially, not temporally. Several models have been proposed to describe a laser beam with a flat-topped profile, such as super Gaussian beam, flattened Gaussian beam, and FMGB [36,37,38,39]. Particularly, the FMGB consists of a sum of Gaussian function components, each with the same spot size and magnitude, in which the spacing between Gaussian functions is chosen to be equal to the spot size.

We propose a new FMGB model to derive the detection models of an RZPN laser altimeter, using the fact that the backscattered waveform is a Gaussian function for a single transmitted beam with a Gaussian shape [27,28,29,30,31]. The full waveform is also a finite sum of Gaussian beams for the transmitted beam with a flat-topped intensity profile in the time domain. In the definition of terminology, the backscattered waveform includes only signals backscattered from the target surface, whereas the received waveform contains signals as well as noise such as background and detector noise. The FMGB model is proposed by

(1)$$FG\left(t\right)=\frac{1}{\beta }\sum_{m=0}^{M}\exp\left[-\frac{{\left(t-mw_{0}\right)}^{2}}{2w_{0}^{2}}\right]$$
with
$$\beta \left(M\right)=\sum_{m=-M}^{M}\exp\left(-\frac{1}{2}m^{2}\right)$$
where M is the FMGB order and $w_{0}$ is the pulse width of the individual Gaussian component. It can be seen from Equation (1) that an FMGB with order M is composed of M+1 Gaussian components. The FMGB reduces to the fundamental Gaussian function when M=0, but it becomes the square or rectangular pulse for M=∞. β is the normalization factor and has a value between unity and 2.5066 for M=0 and M=∞, respectively.

We compute the full width of entire FMGB pulse (i.e., full pulse duration), defining as the time interval at which the amplitude drops down to $1/\sqrt{e}$ with respect to its maximum value. The transmitted pulse modulated with RZPN code has a rectangular shape corresponding to the high order (i.e., large M) in the FMGB. On the high order FMGB composed of many Gaussian beams, the full width, derived in Appendix A, is given by
(2)$$\begin{array}{cc} W_{FG} & \;=Mw_{0}+2\sqrt{2}w_{0}\sqrt{\frac{1}{2}+\ln\beta \left(M\right)}\\ \; & \approx Mw_{0}\;\mathrm{for}\;\mathrm{large}\;M \end{array}$$

Figure 1 indicates that the edge of the entire FMGB pulse becomes sharper when the order gets higher. Consequently, the entire FMGB pulse has a rectangular shape with the full width of $Mw_{0}$ from Equation (2) for a very large M (e.g., $W_{FG}=10\;\mathrm{ns}$ in Figure 1b).

### 2.2. Laser Altimeter with the RZPN Code Modulation Technique

The PN code modulation technique has been widely used for time-resolved measurement applications due to the advantageous property that its cross-correlation has a peak magnitude only when the received sequence matches the bipolar PN code sequence called a kernel. However, it also has several disadvantages; the transmitted laser should be operated at high duty cycle (i.e., 50%) well matched to lasers, resulting in limitation on varying a duty cycle to optimize the system performance, and it is susceptible to background noise and detector dark noise [24,25]. The RZPN code modulation technique is considered to not only overcome these disadvantages but also account for the pulse broadening effect. As shown in Figure 2, the rectangular pulse is transmitted at the beginning of the PN bit interval for a short time and returns to zero for the rest of the bit period, and the kernel has the same values as the PN code kernel but zeros between pulses. 

With the RZPN code modulation and RZ pulse format, the transmitted pulse modulated with a CW laser is written as
(3)$$f_{Tx}\left(t\right)=P_{T}\sum_{n=0}^{N-1}a_{n}u\left(t-nT_{b},T_{p}\right)$$
with
$$u\left(t-nT_{b},T_{p}\right)=\{\begin{array}{cc} 1, & 0\leq t-nT_{b}\leq T_{p}\\ 0, & else \end{array}$$
where $P_{T}$ is the CW laser output power, $a_{n}$ is the PN binary code sequence consisting of one or zero, N is the number of bits in one PN code sequence period, and u(t) is the unit-amplitude function. Substituting the FMGB model with a rectangular shape for the unit-amplitude function to apply the pulse broadening effect in the received waveform, the transmitted pulse is given by
(4)$$f_{Tx}\left(t\right)=P_{T}\sum_{n=0}^{N-1}\frac{a_{n}}{\beta }\sum_{m=0}^{M}\exp\left[-\frac{{\left(t-mw_{0}-nT_{b}\right)}^{2}}{2w_{0}^{2}}\right]$$
where $Mw_{0}$ of the FMGB full width must be equal to the pulse width of the RZPN code modulation.

According to the principle of an RZPN laser altimeter, the RZPN code kernel over one code sequence period is written as
(5)$$\kappa \left(t\right)=\sum_{n=0}^{N-1}{a^{\prime }}_{n}u\left(t-nT_{b},T_{p}\right)$$
with
$${a^{\prime }}_{n}=2a_{n}-1=\{\begin{array}{cc} 1, & a_{n}=1\\ -1, & a_{n}=0 \end{array}$$
where ${a^{\prime }}_{n}$ is the bipolar PN code kernel. From the definitions of unit-amplitude function in Equation (3) and bipolar PN code kernel in Equation (5), the RZPN code kernel has one value among 1, −1, and 0. This allows the cross-correlation of the transmitted RZPN code with its kernel to become near zero when not aligned [24]. This advantage comes from the correlation property of PN code which is given by [5]
(6)$$C_{PN}\left(k\right)=\sum_{i=0}^{N-1}{a^{\prime }}_{i}a_{i+k}=\{\begin{array}{cc} \left(N+1\right)/2 & \mathrm{for}\;k=0\left(\mathrm{mod}N\right)\\ 1/N\approx 0 & \mathrm{for}\;k\neq 0\left(\mathrm{mod}N\right) \end{array}$$

The number of the PN code sequence contains (N+1)/2 ones and (N−1)/2 zeros, which can be generated by a linear feedback shift register consisting of m registers, resulting in $N=2^{m}-1$ [40]. This also provides the balance property that the sum of the bipolar PN codes over one sequence is 1 (i.e., $\sum_{i=0}^{N-1}{a^{\prime }}_{i}=1$). Unless the PN code sequence matches its kernel (i.e., k≠0 (mod N) in Equation (6)), the kernel randomizes the backscattered signal and noise by multiplying them by 1 or −1, and then significantly cancels their effects in the cross-correlation due to the balance property [25].

### 2.3. Received Waveform Models

We assumed that the target surface illuminated by the transmitted pulse consists of small grids with the same reflectivity. The irradiance of laser beam propagation can also be approximated by a Gaussian distribution in the spatial domain, for a fundamental mode laser. For one single Gaussian beam of an FMGB, the backscattered full waveform at the APD input can be written by [35]
(7)$$\begin{array}{cc} f_{RxSG}\left(t\right) & =\sum_{i=1}^{S}p_{RxS,i}\left(t\right)\\ \; & =\sum_{i=1}^{S}\left(\frac{P_{T}}{\pi R_{i}^{2}}\eta_{T}\eta_{R}A_{R}\frac{\rho A_{i}\cos\theta_{i}}{2\pi {\left(z_{i}\tan\theta_{T}\right)}^{2}}\exp\left[-\frac{d_{i}^{2}}{2{\left(z_{i}\tan\theta_{T}\right)}^{2}}\right]\exp\left[-\frac{{\left(t-\tau_{TOF,i}\right)}^{2}}{2w_{0}^{2}}\right]\right) \end{array}$$
where S is the total number of grids of the target surface within the receiver FOV, $p_{RxS,i}$ is the backscattered power by the *i*th grid located at ($x_{i},y_{i}$), $R_{i}$ is the range to the *i*th grid, ρ is the reflectivity of target surface, $A_{i}$ is the *i*th grid area, $\theta_{i}$ is the incident angle on the *i*th grid, $\eta_{T}$ is the transmitter optical efficiency, $\eta_{R}$ is the receiver optical efficiency including the optical filter transmittance, $A_{R}$ is the receiving area, $z_{i}$ is the axial distance to the *i*th grid along the axis of beam propagation, $\theta_{T}$ is the far-field beam divergence half-angle, $d_{i}$ is the radial distance to the *i*th grid from the axis of beam propagation (i.e., $d_{i}^{2}=x_{i}^{2}+y_{i}^{2}$), and $\tau_{TOF,i}$ is the TOF of the *i*th grid. 

Using the theoretical basis that the backscattered waveform can be approximated by a Gaussian function with a pulse width broadened by the target surface profile, the backscattered waveform of Equation (7) can be approximated analytically as
(8)$$f_{RxSG}\left(t\right)=P_{R}\frac{w_{0}}{\;w_{s}}\exp\left[-\frac{{\left(t-\tau_{TOF}\right)}^{2}}{2w_{s}^{2}}\right]$$
with
$$P_{R}=\frac{P_{T}}{\pi R^{2}}\alpha \rho \eta_{T}\eta_{R}A_{R}$$
where $w_{s}$ is the broadened pulse width by the target surface and α is the intensity portion incident on the receiver FOV because the intensity of a laser beam is not uniform in the spatial domain from the axis of beam propagation but distributed with a Gaussian function. The term of $w_{0}/w_{s}$ results from the condition that the energy of the transmitted pulse should be the same as the backscattered pulse. Considering that the laser altimeter is pointed at nadir, the intensity portion can be calculated by
(9)$$\begin{array}{cc} \alpha & =\frac{1}{2\pi {\left(z\tan\theta_{T}\right)}^{2}}\int_{0}^{A_{FOV}}\exp\left(-\frac{x^{2}+y^{2}}{2{\left(z\tan\theta_{T}\right)}^{2}}\right)dA\\ \; & ={\left[erf\left(\frac{\tan\theta_{FOV}}{\sqrt{2}\tan\theta_{T}}\right)\right]}^{2} \end{array}$$
where $A_{FOV}$ is the surface area within the receiver FOV half-angle $\theta_{FOV}$, and erf(⋅) is the error function.

Using Equations (4) and (7), the backscattered full waveform of FMGBs consisting of a finite sum of Gaussian beams can be written by
(10)$$f_{RxMG}\left(t\right)=\sum_{i=1}^{S}\left(\frac{P_{T}}{\pi R_{i}^{2}}\eta_{T}\eta_{R}A_{R}\frac{\rho A_{i}\cos\theta_{i}}{2\pi {\left(z_{i}\tan\theta_{T}\right)}^{2}}\exp\left[-\frac{d_{i}^{2}}{2{\left(z_{i}\tan\theta_{T}\right)}^{2}}\right]\frac{1}{\beta }\sum_{m=0}^{M}\exp\left[-\frac{{\left(t-mw_{0}-\tau_{TOF,i}\right)}^{2}}{2w_{0}^{2}}\right]\right)$$

Applying the theoretical result of Equations (8)–(10), the backscattered analytical waveform can also be approximated as
(11)$$f_{RxMG}\left(t\right)=P_{R}\frac{w_{0}}{\beta \;w_{s}}\sum_{m=0}^{M}\exp\left[-\frac{{\left(t-mw_{0}-\tau_{TOF}\right)}^{2}}{2w_{s}^{2}}\right]$$

For the asteroid surface, full and analytical waveforms of FMGBs (i.e., Equations (10) and (11), respectively) are compared and analyzed by numerical simulations in Section 4. The analytical waveform is achieved from the fact that the backscattered waveform of illuminated spot can be approximated by a Gaussian function as shown Equation (11), whereas the numerical waveform is obtained by combining the backscattered waveforms of individual grid within the illuminated spot as shown in Equation (10).

For the RZPN laser altimeter with a transmitted rectangular pulse (i.e., $Mw_{0}=T_{p}$ for a very large M), the received pulse train at the APD input by using Equations (4) and (11) is written by
(12)$$f_{Rx}\left(t\right)=P_{R}\sum_{n=0}^{N-1}\left(a_{n}\frac{w_{0}}{\beta \;w_{s}}\sum_{m=0}^{M}\exp\left[-\frac{{\left(t-mw_{0}-nT_{b}-\tau_{TOF}\right)}^{2}}{2w_{s}^{2}}\right]\right)$$

Considering the noise sources in Equation (12), the received pulse train at the APD output in terms of multiplied electrons can be expressed as
(13)$$f\left(t\right)=\left(G\frac{\eta_{D}\lambda }{\hbar c}\right)P_{R}f_{Rx}\left(t\right)+\left(G\frac{\eta_{D}\lambda }{\hbar c}\right)b\left(t\right)+w\left(t\right)$$
with
$$\langle b\left(t\right)\rangle =P_{b}=L_{\lambda }\;\Omega_{FoV}\eta_{R}A_{R}\Delta \lambda_{Filter}$$
where G and $\eta_{D}$ are the gain and quantum efficiency of an APD detector, respectively, λ is the wavelength of the transmitted laser, ℏ is the Plank’s constant, c is the light speed, 〈b〉 is the average power of background noise, $L_{\lambda }$ is the background noise irradiance, $\Omega_{FoV}$ is the receiver FOV in steradians, $\Delta \lambda_{Filter}$ is the bandwidth of the optical bandpass filter, and w(t) is the detector noise including dark current and thermal noise. 

### 2.4. Cross-Correlation

An HgCdTe APD detector, operated below the avalanche breakdown voltage for the linear analog output, is considered in this work because it allows nearly quantum-limited detection for weak optical signals by applying a high APD gain without causing additional excess noise and degrading the SNR. The detector dark current consists of three components: surface leakage current, tunnel current, and bulk dark current. The bulk dark current is dominated for a high APD gain because the surface leakage current is not multiplied by the gain, the tunnel current is partially multiplied by the gain, but the bulk dark current is multiplied by the gain [19]. The detector thermal noise is negligible for the HgCdTe detector with a cryo-cooler to keep the detector at cryogenic temperatures [18,25]. Accounting for only the bulk dark current as the detector noise, the mean and variance of multiplied electrons integrated over $\Delta t=t_{2}-t_{1}$ seconds from Equation (13) are given by [41]
(14)$$\langle \int_{t_{1}}^{t_{2}}f\left(t\right)dt\rangle =\left(G\frac{\eta_{D}\lambda }{\hbar c}\right)P_{R}\int_{t_{1}}^{t_{2}}f_{Rx}\left(t\right)dt\;+\left(G\frac{\eta_{D}\lambda }{\hbar c}\right)P_{b}\Delta t+G\frac{I_{b}}{q}\Delta t$$
(15)$$\mathrm{var}\left(\int_{t_{1}}^{t_{2}}f\left(t\right)dt\right)=\left(FG^{2}\frac{\eta_{D}\lambda }{\hbar c}\right)P_{R}\int_{t_{1}}^{t_{2}}f_{Rx}\left(t\right)dt\;+\left(FG^{2}\frac{\eta_{D}\lambda }{\hbar c}\right)P_{b}\Delta t+FG^{2}\frac{I_{b}}{q}\Delta t$$
where $I_{b}$ is the surface dark current, q is the electron charge, and F is the excess noise factor. In Equation (15), the first term ($\sigma_{s}^{2}$) is the variance of signal shot-noise, the second ($\sigma_{bg}^{2}$) and third terms ($\sigma_{d}^{2}$) are the variances of background shot-noise and bulk dark current, respectively.

The cross-correlation between the received pulse train and its kernel, in terms of integration, is defined as
(16)$$\begin{array}{cc} C\left(\tau \right) & =\int_{0}^{T_{c}}f\left(t\right)\kappa \left(t-\tau \right)dt\\ \; & =\left(G\frac{\eta_{D}\lambda }{\hbar c}\right)P_{R}\int_{0}^{T_{c}}f_{Rx}\left(t\right){a^{\prime }}_{n}u_{1}\left(t\right)dt\;+\left(G\frac{\eta_{D}\lambda }{\hbar c}\right)\int_{0}^{T_{c}}b\left(t\right){a^{\prime }}_{n}u_{1}\left(t\right)dt+\int_{0}^{T_{c}}w\left(t\right){a^{\prime }}_{n}u_{1}\left(t\right)dt \end{array}$$
where $u_{1}\left(t\right)$ is a rectangular pulse defined as $u\left(t-nT_{b}-\tau ,T_{p}\right)$ and $T_{c}=NT_{b}$ is the integration time corresponding to the code length period. The time shift defined as $\tau =kT_{b}+\Delta \tau$ is introduced to apply the advantageous correlation property of Equation (6), where k is the same integer as Equation (6) and $0\leq \Delta \tau <T_{b}$. Using the definition of $\tau_{w}$ as shown in Figure 2, which is the TOF corresponding to a grid with the shortest range within the receiver FOV, the cross-correlation can be rewritten by
(17)$$\begin{array}{cc} C\left(k,\Delta \tau \right) & =\left(G\frac{\eta_{D}\lambda }{\hbar c}\right)P_{R}\int_{\tau_{w}}^{\tau_{w}+T_{c}}\sum_{n=0}^{N-1}a_{n}{a^{\prime }}_{n+k}\frac{w_{0}}{\beta \;w_{s}}\sum_{m=0}^{M}\exp\left[-\frac{{\left(t-mw_{0}-nT_{b}-\tau_{TOF}\right)}^{2}}{2w_{s}^{2}}\right]u_{2}\left(t\right)dt\\ \; & +\left(G\frac{\eta_{D}\lambda }{\hbar c}\right)\int_{\tau_{w}}^{\tau_{w}+T_{c}}b\left(\tau \right){a^{\prime }}_{n+k}u_{2}\left(t\right)d\tau +\int_{\tau_{w}}^{\tau_{w}+T_{c}}w\left(\tau \right){a^{\prime }}_{n+k}u_{2}\left(t\right)d\tau \end{array}$$
where $u_{2}\left(t\right)$ is a rectangular pulse defined as, $u_{2}\left(t\right)=u\left(t-nT_{b}-\tau_{w}-\Delta \tau ,T_{p}\right)$. The bit interval is chosen so that the received full waveform exists within the bit interval to improve the performance of the RZPN laser altimeter. Using the balance and correlation properties as well as definition of RZPN code kernel, with some mathematical effort and Equation (14), the expected cross-correlation can be written by
(18)$$\begin{array}{cc} \langle C\left(k=0,\Delta \tau \right)\rangle = & \left(G\frac{\eta_{D}\lambda }{\hbar c}\right)P_{R}\frac{N+1}{2}\frac{w_{0}}{\beta w_{s}}\int_{\tau_{w}+\Delta \tau }^{\tau_{w}+\Delta \tau +T_{p}}\sum_{m=0}^{M}\exp\left[-\frac{{\left(t-mw_{0}-\tau_{TOF}\right)}^{2}}{2w_{s}^{2}}\right]dt\\ \; & +\left(G\frac{\eta_{D}\lambda }{\hbar c}\right)P_{b}T_{p}+G\frac{I_{b}}{q}T_{p} \end{array}$$
(19)$$\langle C\left(k\neq 0,\Delta \tau \right)\rangle =\mu_{b}=\left(G\frac{\eta_{D}\lambda }{\hbar c}\right)P_{b}T_{p}+G\frac{I_{b}}{q}T_{p}$$

Equation (19) corresponds to the minimum cross-correlation, independently of k and Δτ if and only if k≠0. As shown in Figure 2, the cross-correlation has a maximum value at k=0 and $\Delta \tau =\tau_{TOF}-\tau_{w}$. As derived in Appendix B, the maximum cross-correlation depending on the transmitted and broadened pulse width is achievable in the analytical form as
(20)$$\begin{array}{cc} \langle C_{\max}\rangle =\;\mu_{p} & =\;\left(G\frac{\eta_{D}\lambda }{\hbar c}\right)P_{R}\frac{N+1}{2}T_{p}\xi_{TOF}\left(w_{s},T_{p}\right)+\left(G\frac{\eta_{D}\lambda }{\hbar c}\right)P_{b}T_{p}+G\frac{I_{b}}{q}T_{p}\\ \; & \approx \left(G\frac{\eta_{D}\lambda }{\hbar c}\right)P_{R}\frac{N+1}{2}T_{p}\xi_{TOF}\left(w_{s},T_{p}\right) \end{array}$$
with
$$\xi_{TOF}\left(w_{s},T_{p}\right)=\frac{1}{\beta }\left[\sqrt{2\pi }erf\left(\frac{T_{p}}{\sqrt{2}w_{s}}\right)+2\frac{w_{s}}{T_{p}}\exp\left(-{\left(\frac{T_{p}}{\sqrt{2}w_{s}}\right)}^{2}\right)-2\frac{w_{s}}{T_{p}}\right]$$
where $\xi_{TOF}\left(w_{s},T_{p}\right)$ means the pulse broadening effect on the maximum cross-correlation of the RZPN laser altimeter, whose characteristics are analyzed by the numerical simulation. The last two terms in Equation (20) are negligible due to the large number of bits and small transmitted pulse width (e.g., N=127 and $T_{p}=8\;\mathrm{ns}$ in this study). From the property of $\xi_{TOF}\left(w_{s},T_{p}\right)\leq 1$, $\xi_{TOF}\left(w_{s},T_{p}\right)=1$ shows that the beam divergence angle is extremely small, and consequently, the received pulse is not broadened.

## 3. Performance Metrics

### 3.1. Signal-to-Noise Ratio

SNR is an important figure to determine the ranging performance of range-resolved systems with code modulation techniques because the target distance is calculated from a peak search of the cross-correlation. It is necessary to accumulate the received pulse trains into a histogram (i.e., accumulation over a number L of received pulse trains) to achieve a high value of the peak cross-correlation and improve SNR by $\sqrt{L}$ times. SNR is defined, in terms of multiplied electrons, as the mean value to the standard deviation ($\sigma_{p}$) for the maximum cross-correlation:(21)$$\mathrm{SNR}=\frac{L\cdot \langle C_{\max}\rangle }{\sqrt{L\cdot \mathrm{var}\left(C_{\max}\right)}}=\sqrt{L}\frac{\mu_{p}}{\sigma_{p}}$$

It is worth noting from Equation (15) that the variance of multiplied electrons over the fixed integration interval is dependent on the start integration time because the received waveform for a transmitted rectangular pulse is a time-dependent function, as shown in Equation (11). In consequence, the variance of cross-correlation is dependent on a time shift of the RZPN code kernel; in other words, dependent of Δτ regardless of k in Equation (18) because of the property of variance (i.e., $\mathrm{var}\left(\mathrm{aX}\right)=a^{2}\mathrm{var}\left(X\right)$ where a and X are a constant and a random variable, respectively). Applying Equation (18) into the property of Equation (15) and using the central limit theorem, the variance of cross-correlation can be written by
(22)$$\mathrm{var}\left(C\left(\tau \right)\right)=\left(FG^{2}\frac{\eta_{D}\lambda }{\hbar c}\right)P_{R}\frac{N+1}{2}T_{p}\xi \left(w_{s},T_{p},\tau \right)\;+NFG^{2}\left(\frac{\eta_{D}\lambda }{\hbar c}P_{b}+\frac{I_{b}}{q}\right)T_{p}$$
where $\xi \left(w_{s},T_{p},\tau \right)$ is defined in Appendix B and ranges from 0 to $\xi_{TOF}\left(w_{s},T_{p},\tau \right)$. $\xi \left(w_{s},T_{p},\tau \right)=0$ indicates that the integration interval of $T_{p}$ exists outside the received waveform corresponding to the transmitted rectangular pulse and before the next received waveform, in other words, the integration is performed in the region of no received signal. Therefore, the maximum and minimum variances of cross-correlation are given by
(23)$${\mathrm{var}}_{\max}=\left(FG^{2}\frac{\eta_{D}\lambda }{\hbar c}\right)P_{R}\frac{\left(N+1\right)}{2\;}T_{p}\xi_{TOF}\left(w_{s},T_{p}\right)+NT_{p}FG^{2}\left(\frac{\eta_{D}\lambda }{\hbar c}P_{b}+\frac{I_{b}}{q}\right)$$
(24)$${\mathrm{var}}_{\min}=NT_{p}FG^{2}\left(\frac{\eta_{D}\lambda }{\hbar c}P_{b}+\frac{I_{b}}{q}\right)$$

The maximum variance of cross-correlation happens at $\Delta \tau =\tau_{TOF}-\tau_{w}$ in Equation (17) regardless of k=0 or k≠0. In particular, we define $\sigma_{p}^{2}={\mathrm{var}}_{\max}$ when k=0 and $\sigma_{b}^{2}={\mathrm{var}}_{\max}$ when k≠0. When $\Delta \tau >T_{w}$ and the bit interval is designed to be so large that $T_{b}>T_{w}+T_{p}$, the minimum variance of cross-correlation happens. It means that the backscattered pulse train has absolutely no effect on the cross-correlation by the return-to-zero property of the RZPN kernel.

When considering the signal shot-noise limited detection (i.e., $\sigma_{s}>>\sigma_{bg},\sigma_{d}$), the SNR can be approximated by
(25)$$\mathrm{SNR}=\sqrt{L}\frac{\left(G\frac{\eta_{D}\lambda }{\hbar c}\right)P_{R}\frac{N+1}{2}T_{p}\xi_{TOF}\left(w_{s},T_{p}\right)}{\sqrt{\left(FG^{2}\frac{\eta_{D}\lambda }{\hbar c}\right)P_{R}\frac{N+1}{2}T_{p}\xi_{TOF}\left(w_{s},T_{p}\right)}}={\mathrm{SNR}}_{0}\sqrt{\xi_{TOF}\left(w_{s},T_{p}\right)}$$
with
$${\mathrm{SNR}}_{0}=\frac{\sqrt{L}}{\sqrt{F}}\sqrt{\left(\frac{\eta_{D}\lambda }{\hbar c}\right)P_{R}\frac{N+1}{2}T_{p}}$$
where ${\mathrm{SNR}}_{0}$ is the SNR when the received pulse is not broadened. As seen in Equation (25), the SNR is proportional to the square root of the pulse broadening effect of $\xi \left(w_{s},T_{p}\right)$. However, $\xi_{TOF}\left(w_{s},T_{p}\right)$ becomes smaller according to the broadened pulse width and consequently degrades the ranging performance by decreasing the SNR. The large accumulation number (L) and long code sequence length ($NT_{b}$) not only improve the ranging performance but also decrease the detection speed. This will make a laser altimeter unsuitable for high dynamic targets because accumulated pulse trains are shifted in the histogram due to the line-of-sight velocity of targets.

### 3.2. Detection Probability

The detection probability is also a significant parameter to determine the ranging performance to targets, along with the SNR. The target detection is performed by searching a peak of cross-correlation which may not match with the target distance due to a random peak of cross-correlation by Gaussian noise, corresponding to a cross-correlation with $\mu_{b}$ and $\sigma_{b}$. The detection probability is expressed in an integration form as [25]
(26)$$P_{D}={\left[\frac{1}{2\pi \sigma_{p}\sigma_{b}}\int_{-\infty }^{\infty }\exp\left(-\frac{{\left(z_{p}-\mu_{p}\right)}^{2}}{2\sigma_{p}^{2}}\right)\int_{-\infty }^{z_{p}}\exp\left(-\frac{{\left(z_{b}-\mu_{b}\right)}^{2}}{2\sigma_{b}^{2}}\right)dz_{b}dz_{p}\right]}^{N-1}$$

Applying a *Q*-function and a Gauss–Hermite quadrature integration to achieve a closed-form expression, Equation (26) can be rewritten by [22]
(27)$$P_{D}=\frac{1}{\sqrt{\pi }}\sum_{i=1}^{n}w_{i}Q\left(-\frac{\sqrt{2}\sigma_{p}x_{i}+\mu_{p}-\mu_{b}}{\sigma_{b}}\right)$$
where Q(⋅) is the *Q*-function referred to as the Gaussian probability integral and $x_{i}$ and $w_{i}$ are the zero points and weight factors for the nth-order Hermite polynomial, respectively. The order of the Hermite polynomial is chosen so that the desired degree of accuracy is achieved. Considering the accumulation number, the substitution of Equations (19), (20) and (22) into Equation (27) leads to
(28)$$\begin{array}{cc} P_{D} & =\frac{1}{\sqrt{\pi }}\sum_{i=1}^{n}w_{i}Q\left(-\sqrt{2}x_{i}-SNR\right)\\ \; & =\frac{1}{\sqrt{\pi }}\sum_{i=1}^{n}w_{i}Q\left(-\sqrt{2}x_{i}-SNR_{0}\sqrt{\xi_{TOF}\left(w_{s},T_{p}\right)}\right) \end{array}$$

The detection probability also reduces as the pulse broadening effect increases (i.e., $\xi_{TOF}\left(w_{s},T_{p}\right)$ becomes smaller) because a *Q* function exhibits rapid exponential decay.

## 4. Numerical Results 

In this section, we present the numerical results not only to validate the analytical waveform model of the backscattered signal for a transmitted rectangular pulse, but also to investigate the pulse broadening effect on the cross-correlation and the ranging performance. As shown in Figure 3, two spots on asteroid 25143 Itokawa were considered as the target surfaces, whose three-dimensional model is available from NASA resources (http://nasa3d.arc.nasa.gov, accessed on 1 November 2021). The numerical simulation was performed for the reconnaissance or mapping mode in which the line-of-sight velocity is small (i.e., from the order of centimeters to meters per second). This mode, in turn, enables the histogram.

Itokawa is classified as an S-type asteroid, and its reflectivity is ρ=0.28±0.02 from ground-based mid-infrared photometric observations and infrared astronomical satellite measurements [42]. The background noise irradiance is $L_{\lambda }=0.27$ W/m^2^∙sr∙nm at a 1550 nm wavelength by assuming Itokawa is one astronomical unit away from the Sun. The PN code sequence is generated by a linear feedback shift register with seven stages, providing $N=2^{7}-1=127$ bits, $T_{b}=40$ ns of bit interval, and $T_{p}=8$ ns of the pulse width which is implemented with M=800 of the FMGB order and $w_{0}=10$ ps of individual pulse width (i.e., $T_{p}=800\times 10$ ps). Applying the accumulation number of L=2000 to improve the ranging performance, the total integration time was about 1 ms (~1 kHz) resulting from the equation of $L\times T_{b}\times N$. An HgCdTe APD detector with a cryo-cooler is used to allow for single photon sensitivity in the linear analog mode. Considering a CW laser diode based on the MOPA architecture, the transmitter and receiver parameters are listed in Table 1.

The backscattered pulse width is proportional to the range to the target surface because the surface area within the receiver FOV becomes larger depending on the range. Therefore, two locations at two individual spots (i.e., four locations total) were simulated to investigate the pulse broadening effect on the performance of an RZPN laser altimeter. Figure 4 shows the backscattered full waveforms of impulse response and fitting functions at the four locations, in which the full waveform was achieved from Equation (7) with the assumption that the target surface consisted of small grids. At the heights of z=10 km and z=50 km, the spot sizes within the receiver FOV were about 60 × 60 cm^2^ and 300 × 300 cm^2^, the total number of grids was 2542 and 65,007 for spot A, and 3059 and 73,045 for spot B. The area of individual grid was smaller than 20 cm^2^. As shown in Figure 4, the broadened pulse width ($w_{s}$) was estimated by employing a Gaussian fitting to be 0.26 and 1.27 ns for spot A and 0.7 and 1.58 ns for spot B, at the heights of z=10 km and z=50 km, respectively. As expected from the topographic maps shown in Figure 3, the surface profile of spot B made the backscattered pulse width larger than spot A, resulting from a steeper surface slope and larger roughness.

For a transmitted rectangular pulse, its backscattered waveform was demonstrated in terms of the full waveform and analytical waveform obtained from Equations (10) and (11), respectively. As shown in Figure 5a–c, the full waveform could be approximated accurately by the analytical waveform based on the FMGB model, for a smooth surface profile corresponding to Figure 4a,c or the narrow pulse width of impulse response corresponding to Figure 4a,b. However, Figure 5d shows that the analytical waveform was a little different from the full waveform due to the rough surface profile as well as the long range to the target surface. Compared to the waveform in Figure 4, the smooth waveforms in Figure 5 resulted from the superposition of many backscattered waveforms with the minute time interval of $w_{0}$ given in Equation (11). The backscattered wave form became broader by the target surface profile and spot size due to the beam divergence. Even though the pulse width of transmitted rectangular beam was $T_{p}=8$ ns, the backscattered waveform width (i.e., $T_{w}$ in Figure 2) was about 9.4 and 14.6 ns for spot A, 10.2 and 16.2 ns for spot B, at the heights of z=10 km and z=50 km, respectively. The shape and width of backscattered waveform characterized not only the figure of cross-correlation as shown in Figure 6 and Figure 7, but also had an effect on the ranging performance.

For the numerical simulation, noise was modeled by Gaussian random variables and generated at each sampling time of 100 ps. The upper part in Figure 6a shows that the PN code sequence had an almost uniform distribution of zeros and ones, resulting in balance and correlation properties. The received waveforms in Figure 6b and Figure 7b were different from the pattern of the PN code sequence because the backscattered signal was weaker than random noise including background shot noise and detector noise. When the backscattered signal was strong and caused large signal shot noise (signal shot-noise limited case in Figure 7a), the received waveform fluctuated but showed a similar pattern of PN code sequence, resulting in a high value of peak cross-correlation.

In the absence of pulse broadening effect and noise, the cross-correlation should have a shape of perfect triangle, and its width (i.e., $2T_{w}$ in Figure 2) should also be the same as two times of the transmitted pulse width (i.e., 16 ns in the simulation). Regardless of the full or analytical waveform, the peak value and width of cross-correlation were 4.3 × 10^6^ and 19.6 ns in Figure 6d, 9.6 × 10^6^ and 16.8 ns in Figure 6e and Figure 7e. However, the cross-correlation in Figure 7d had two peak values of 4.2 × 10^6^ and 4.3 × 10^6^, and two width values of 21.2 and 23.4 ns for the full and analytical waveforms, respectively. These values of cross-correlation width were larger than $2T_{p}$ due to the pulse broadening effect, but apparently smaller than $2T_{w}$ obtained from Figure 5 due to the variation of signal and noise at the APD output. For the peak value and width of cross-correlation in Figure 7d, the difference between the full and analytical waveforms resulted from that the Gaussian fitting function was not matched well to the backscattered waveform as shown in Figure 5d.

As shown in Figure 6c–e and Figure 7e, the analytical waveform provided the same cross-correlation function as the full waveform for a smooth surface profile (e.g., spot A) or the narrow pulse width of impulse response (e.g., z=10 km at spot A and B). Figure 7d shows that the cross-correlation using the full waveform was slightly shifted to the left compared to the analytical waveform. This shift was caused by the shape difference between the backscattered analytical and full waveform, as shown in Figure 5d. Figure 6c–e and Figure 7c–e show that the cross-correlation function near the mean TOF became broader in shape and rounder at the peak cross-correlation as the pulse width of impulse response increased. This led to the uncertainty of the accurate location of peak cross-correlation and degradation of ranging accuracy. It is demonstrated from Figure 6 and Figure 7 that the analytical waveform was similar to the full waveform obtained from the numerical simulation which subsequently provided the valuable results of the cross-correlation for the analysis of the ranging performance.

The pulse broadening effect factor ($\xi_{TOF}$) plays a significant role in terms of the analysis of SNR as well as detection probability, which is affected by a target surface profile. As shown in Figure 8a, $\xi_{TOF}$ was inversely proportional to the pulse width of the backscattered waveform, but in proportion to the transmitted pulse width. For a fixed pulse width, Equation (20) shows that the maximum cross-correlation can be reduced when $\xi_{TOF}$ decreases (i.e., when the backscattered pulse becomes broader), which indicates that the ranging performance is deteriorated by a target surface profile. The individual component of $\xi_{TOF}$ was analyzed for more examination. As shown in Figure 8b, the first term ($\xi_{1}$) was dominant in the region of the small broadened pulse due to the properties of error function, whereas the effect of the second and third terms ($\xi_{2}$, $\xi_{3}$) increased in the opposite direction as the pulse of the backscattered waveform became wider. 

For the small beam divergence of $\theta_{T}=30$ µrad, the surface roughness was dominant in the backscattered pulse width rather than the range. Therefore, the backscattered pulse width was assumed to be independent of the operation range to investigate SNR in terms of the broadened pulse width and operation range. As shown in Figure 9a, SNR decreased exponentially as the operation range increased because the received power was inversely proportional to the square of range. At the maximum cross-correlation, the broadened pulse width reduced the number of multiplied electrons from the backscattered signal within the integration interval of $T_{p}=8$ ns and consequently also reduced SNR. For example, SNR was 136.5 and 77.1 at the operation range of 10 km, and 4.6 and 2.0 at 100 km, for $w_{s}=0$ ns (i.e., $\xi_{TOF}=1$) and $w_{s}=8$ ns (i.e., $\xi_{TOF}=0.37$), respectively. It is demonstrated from Figure 9a that the broadened pulse width significantly dropped SNR and results in the deterioration of the ranging precision which was inversely proportional to SNR at a fixed sampling time [25].

The standard deviations of background shot-noise and bulk dark current were constant, independently of the operation range, with $\sigma_{bg}=12.2$ and $\sigma_{d}=2.8\times 10^{4}$. However, the standard deviation of signal shot-noise, from Equation (15), depended on the received power (i.e., operation range), with $\sigma_{s}=8.6\times 10^{4}$ and $\sigma_{s}=5.2\times 10^{4}$ at the operation range of 10 km and $\sigma_{s}=8.6\times 10^{3}$ and $\sigma_{s}=5.2\times 10^{3}$ at 100 km, for $w_{s}=0$ and $w_{s}=8$ ns, respectively. As shown in Figure 9b, SNR was similar to $SNR_{shot}$ at a short range; however, $SNR_{shot}$ was much larger than SNR at a long range. The difference between SNR and $SNR_{shot}$ increased as the backscattered pulse width increased, particularly at a long range. This indicates that SNR was dominated by the signal shot-noise at a short range and by the bulk dark current at a long range, and the pulse broadening effect had a greater effect on SNR under a weak backscattered signal.

Figure 10 shows the detection probability in terms of the operation range, in which the detection probability rapidly decreased as the operation range increased and the broadened pulse width became broader. It is shown from Equation (28) that SNR should be larger than 10.36 to achieve a detection probability of 100%. From the numerical simulation, the maximum operational ranges to satisfy the detection probability of 100% were 63.2, 49.3, 38.4, and 27.9 km for the broadened pulse width of $w_{s}=$ 0, 4, 8, and 16 ns which may occur at the long operation range, respectively. The pulse broadening effect significantly limited the operational range for the reconnaissance and mapping modes. The reconnaissance mode means a coarse ranging survey from a long distance for the shape and rotation axis of an asteroid, whereas the mapping mode provides precise ranging measurements from a mid-altitude orbit for a global geodetic model and scientific investigations [26]. In the long-range reconnaissance mode, the backscattered pulse width is broadened by not only the surface slope and roughness but also a large footprint due to the beam divergence. Thus, the pulse broadening effect plays a critical role in degrading the detection probability as well as SNR for laser altimeters using the code modulation technique.

## 5. Conclusions

Code modulation techniques permit laser altimeters to be compact, robust, and power efficient with long lifetime for space-based applications. However, the backscattered pulse is broadened by the beam divergence as well as target surface profiles, and then deteriorates the ranging performance in terms of SNR and detection probability by bringing down the maximum cross-correlation. Taking into consideration that the impulse response is approximated by a Gaussian function, an analytical waveform of backscattered pulses was derived using a new FMGB model, for the transmitted rectangular pulse. Closed-form expressions were also achievable using the analytical waveform in terms of the pulse broadening effect for the SNR and detection probability as well as the cross-correlation.

Asteroid 25143 Itokawa was considered for more practical investigation in the numerical simulations, on which four target surface profiles were chosen at two individual heights for two spots, respectively, providing the backscattered pulse width in the range of 0.27 to 1.58 ns. It was demonstrated that the full waveform could be approximated accurately by the analytical waveform, and that the cross-correlation functions were the same between the full and analytical waveforms under conditions of smooth target surface profiles or a narrow pulse width of impulse response. As expected from closed-form expressions with the property of $\xi_{TOF}\left(w_{s},T_{p}\right)\leq 1$, we also demonstrated that the pulse broadening effect significantly reduced the ranging performance of SNR and detection probability. As the broadened pulse width increased from 0 to 8 ns, SNR dropped from 136.5 to 77.1 at the operation range of 10 km, and the maximum operation range with the detection probability of 100% decreased significantly from 63.2 to 38.4 km. Thus, the pulse broadening effect is a critical design parameter which should be considered when designing laser altimeters using the code modulation technique and when evaluating their ranging performance.

## Figures and Tables

**Figure 1 sensors-22-03293-f001:**
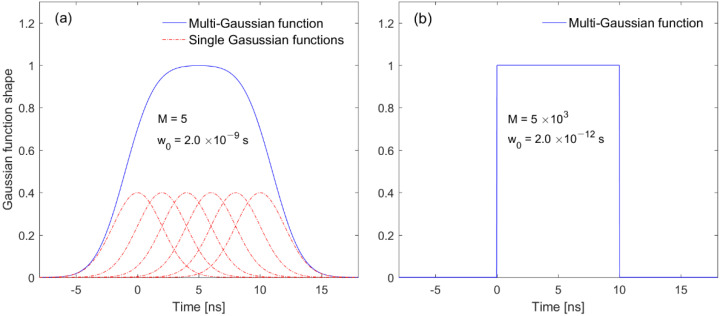
Pulse shapes of the entire FMGB depending on the orders. Pulse shapes for M=5 (**a**) and M=5000 (**b**), respectively.

**Figure 2 sensors-22-03293-f002:**
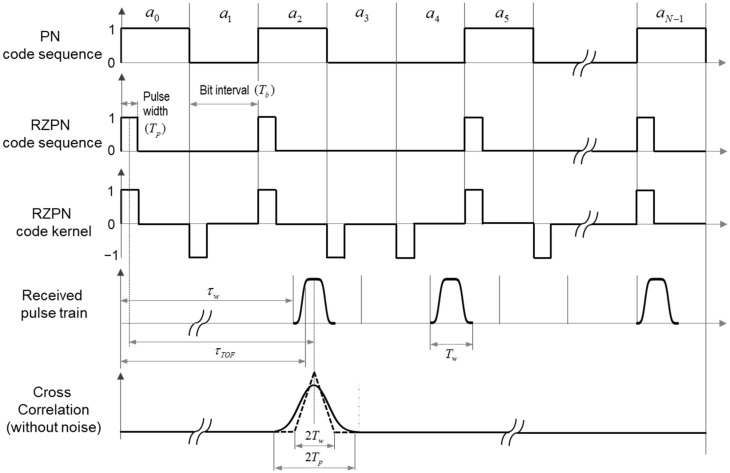
Illustration of RZPN laser altimeter principle. $\tau_{TOF}$ representing the mean TOF to the target surface indicates the time difference between the centers of the transmitted rectangular pulse and corresponding received pulse. $\tau_{w}$ is the shortest TOF for the first transmitted pulse corresponding to m=0 in the FMGB model, and $T_{w}$ is the time difference between the shortest TOF for the first transmitted pulse and the longest TOF for the last transmitted pulse corresponding to m=M. For the cross-correlation in absence of noise, the solid line means the cross-correlation of broadened pulse trains, whereas the dotted line with a triangle shape corresponds to the case in which the received pulse is not broadened.

**Figure 3 sensors-22-03293-f003:**
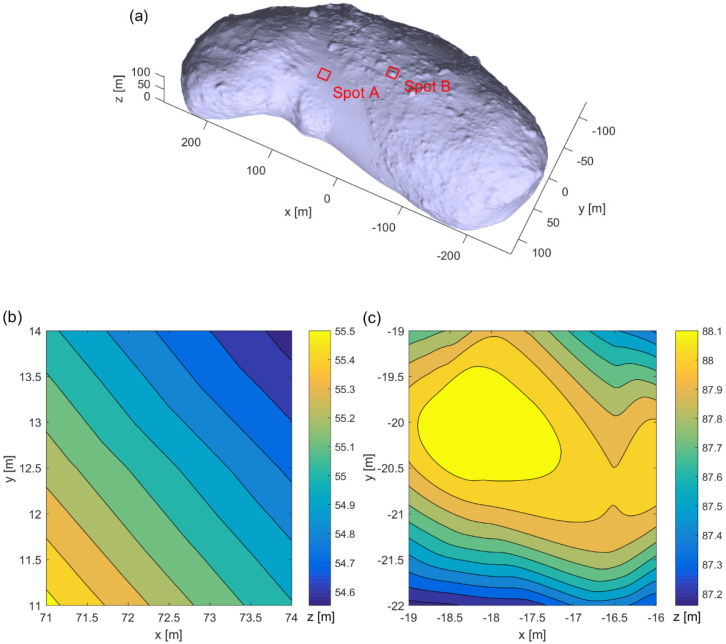
Two spots in the three-dimensional model of Itokawa (**a**) and topographic maps showing land contours of spot A (**b**) and spot B (**c**). The topographic map shows that the two spots are located at the slope face and small hill, respectively.

**Figure 4 sensors-22-03293-f004:**
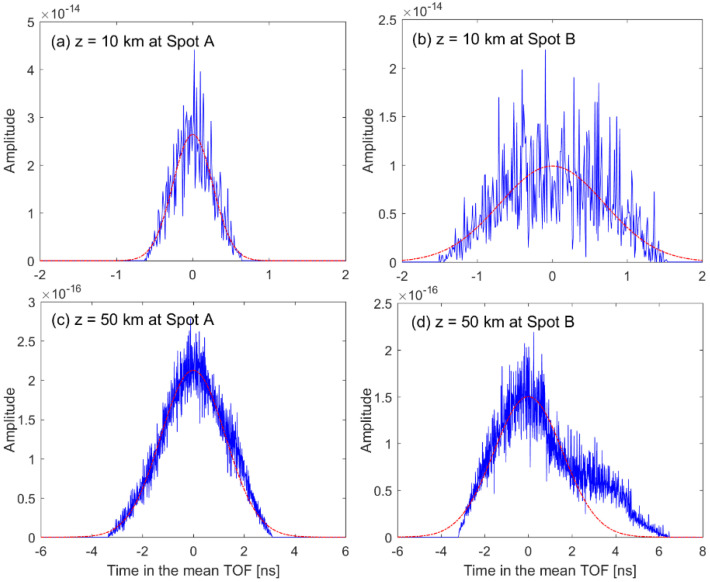
Backscattered waveforms of impulse response at the height of z=10 km and z=50 km for spot A and spot B, respectively, and their fitting functions. The blue solid line represents the impulse response, and the red dash-dotted line corresponds to the fitting function which indicates the broadened pulse width.

**Figure 5 sensors-22-03293-f005:**
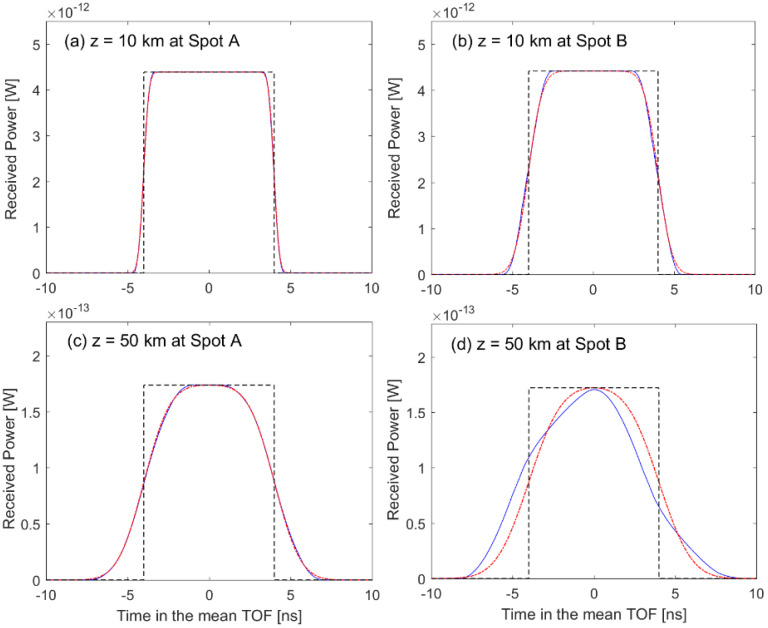
Backscattered waveform for a transmitted rectangular pulse at four locations. The blue solid line represents the full waveform, the red dash-dotted line represents the analytical waveform to apply the broadened pulse width estimated from a Gaussian fitting, and the black dashed line corresponds to the transmitted pulse width.

**Figure 6 sensors-22-03293-f006:**
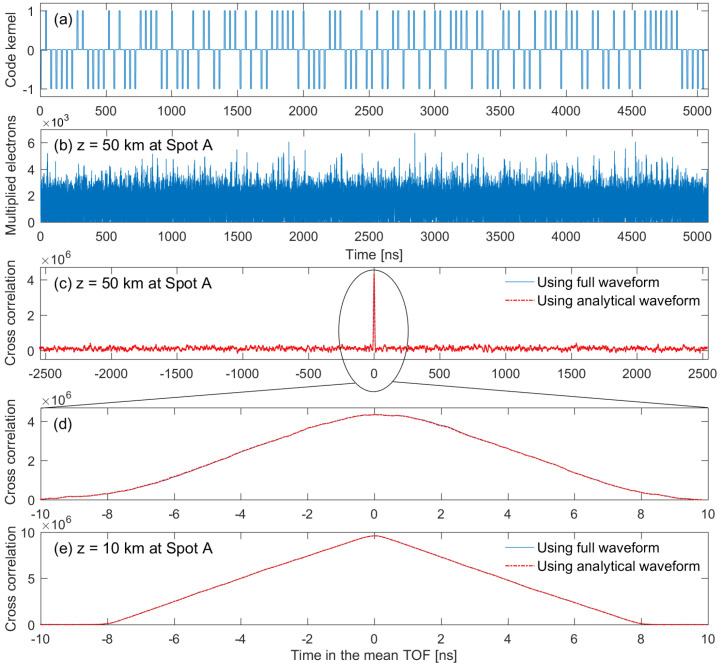
RZPN code kernel (**a**), received full waveform (**b**), and cross-correlations (**c**–**e**) using the full waveform and analytical waveform for spot A. (**d**,**e**) show the results of cross-correlation over a finer time scale to investigate the characteristics of the cross-correlation shape.

**Figure 7 sensors-22-03293-f007:**
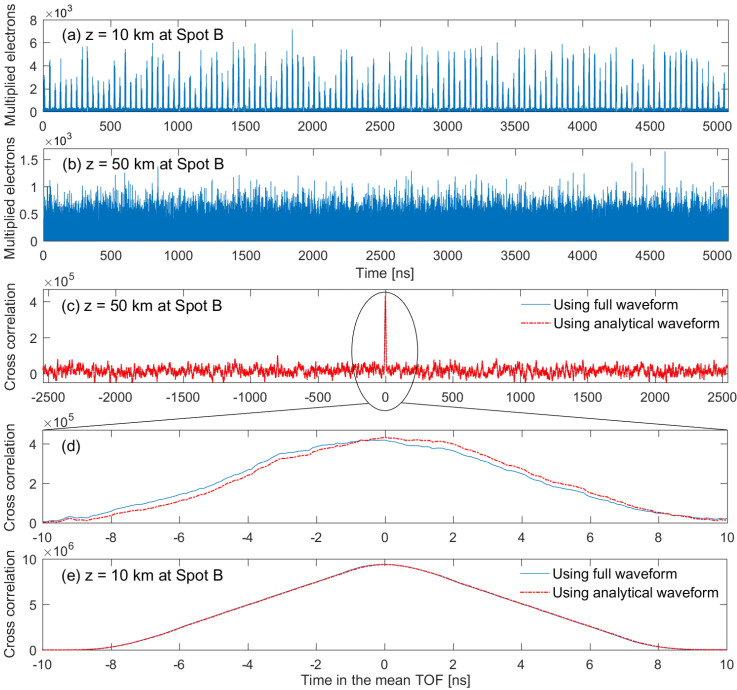
Full waveforms (**a**,**b**) and cross-correlations (**c**–**e**) using the full waveform and analytical waveform for spot B. (**d**,**e**) show the results of cross-correlation over a finer time scale to investigate the characteristics of cross-correlation shape by comparison with Figure 6.

**Figure 8 sensors-22-03293-f008:**
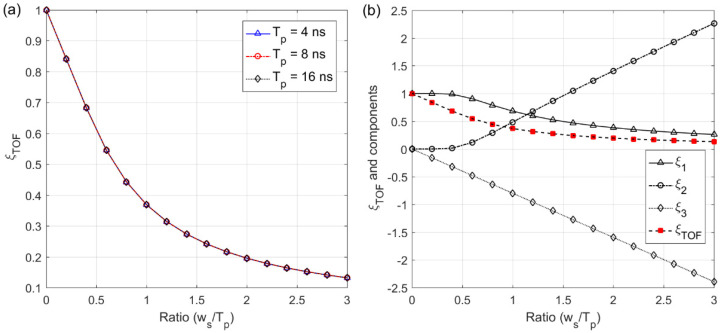
Pulse broadening effect factor at the maximum cross-correlation (**a**) and its individual component values (**b**) at $T_{p}=8$ ns. The components of $\xi_{1}$, $\xi_{2}$, and $\xi_{3}$ correspond to the first, second and third terms in the definition of the pulse broadening effect factor given in Equation (19).

**Figure 9 sensors-22-03293-f009:**
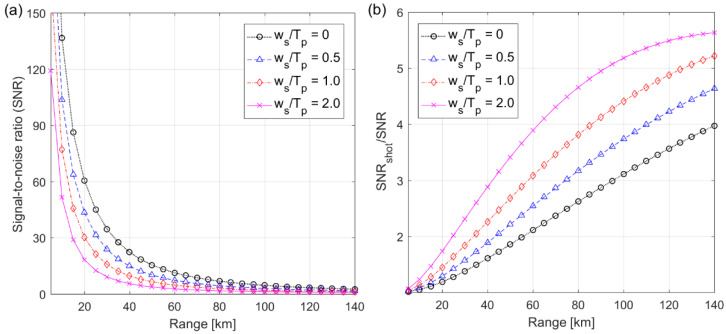
Signal-to-noise ratio (**a**) and ratio of signal-to-noise ratio of the signal shot-noise limited detection to the noise detection (**b**) for the fixed pulse width of $T_{p}=8$ ns. $SNR_{shot}$ is the signal-to-noise ratio corresponding to the signal shot-noise limited detection to neglect background noise and detector noise.

**Figure 10 sensors-22-03293-f010:**
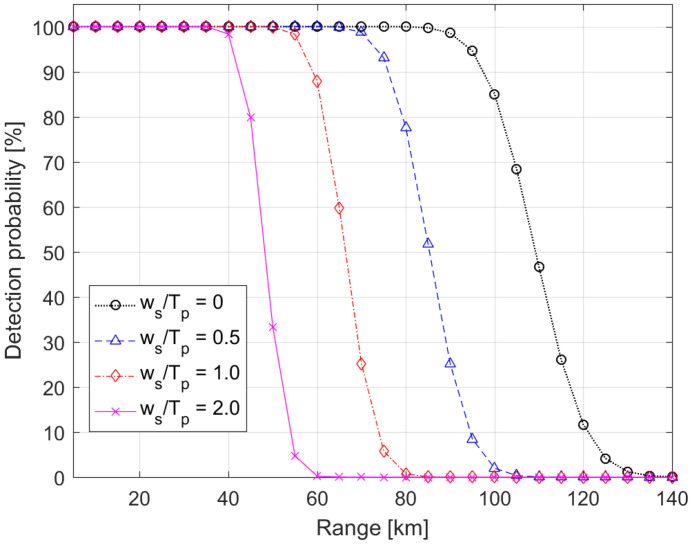
Detection probability for the fixed pulse width of $T_{p}=8$ ns.

**Table 1 sensors-22-03293-t001:** Transmitter and receiver parameters used in the numerical simulation.

Parameter Description	Symbol	Value
Laser wavelength	λ	1550 nm
Average laser power	$P_{T}$	2 W
Tx optical efficiency	$\eta_{T}$	0.90
Effective Rx area	$A_{R}$	78.5 cm^2^
Rx optical efficiency	$\eta_{R}$	0.75
Beam divergence	$\theta_{T}$	30 µrad
Receiver FOV	$\theta_{FOV}$	30 µrad
Optical filter bandwidth	$\Delta \lambda_{Filter}$	10 nm
APD quantum efficiency	$\eta_{D}$	0.7
APD gain	G	500
APD excess noise factor	F	1.2
Bulk dark current	$I_{b}$	0.2 pA

## Data Availability

Data sharing not applicable.

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
