# Peer review of "Pulse Broadening Effects on Ranging Performance of a Laser Altimeter with Return-to-Zero Pseudorandom Noise Code Modulation"

_sensors, 2022, doi:10.3390/s22093293_

Round 1

Reviewer 1 Report

This is a very interesting study. It also has some inspiration for the design of higher precision laser altimeter. This paper analyzes the influence of pulse brordening effects on laser ranging from two aspects: signal-to-noise ratio and observation probability. The results of the study have great significance for the laser altimeter.

The slight deficiency is that there are few practical examples to verify the mathematical model. Sufficient quantitative analysis also needs to be supplemented. The explanations of figures 6 and 7 are not clear enough. It is necessary for the design of new load that the simulation analysis is done well enough. Therefore, the more thorough the analysis of simulation results, the higher the feasibility of the results.

In the process of writing, it is better to change some sentences in the active voice to the passive voice. After all, this is a scientific paper.

Reviewer 2 Report

Please see the attached document for general and specific comments about this manuscript.

Round 2

Reviewer 2 Report

Thank you for your careful responses to the review and changes made to the manuscript. This work is now suitable for publication right away.